# Proteomic Analysis Identifies Dysregulated Proteins in Albuminuria: A South African Pilot Study

**DOI:** 10.3390/biology13090680

**Published:** 2024-08-30

**Authors:** Siyabonga Khoza, Jaya A. George, Previn Naicker, Stoyan H. Stoychev, June Fabian, Ireshyn S. Govender

**Affiliations:** 1Department of Chemical Pathology, National Health Laboratory Service and School of Pathology, Faculty of Health Sciences, University of the Witwatersrand, Johannesburg 2000, South Africa; 2Wits Diagnostic Innovation Hub, University of the Witwatersrand, Johannesburg 2000, South Africa; 3Future Production Chemicals, Council for Scientific and Industrial Research, Pretoria 0001, South Africa; 4ReSyn BioSciences, Edenvale 1610, South Africa; 5Evosep Biosystems, 5230 Odense, Denmark; 6Wits Donald Gordon Medical Centre, School of Clinical Medicine, Faculty of Health Sciences, University of the Witwatersrand, Johannesburg 2000, South Africa; 7Medical Research Council/Wits University Rural Public Health and Health Transitions Research Unit (Agincourt), School of Public Health, Faculty of Health Sciences, University of the Witwatersrand, Johannesburg 2000, South Africa

**Keywords:** urinary proteomics, albuminuria, chronic kidney disease, biomarker

## Abstract

**Simple Summary:**

Chronic kidney disease remains a global health priority, only detected at relatively advanced stages by current markers. Identifying alternative markers for early detection is imperative. In this study, we profiled the urinary proteome in patients with albuminuria and well-preserved eGFR. We identified 80 proteins that were differentially abundant between the cases (albuminuria) and controls (normoalbuminuria). Among these, 12 proteins (SERPINA1, ALB, SERPINC1, AFM, PIGR, A1BG, COL6A1, MYG, LV39, MUC1, ICOSLG, and UMOD) had the highest discriminating abilities (area under curve > 0.8) between the cases and controls. When differentially abundant proteins were combined into an 80-protein model, the model was able to predict cases from controls with a predictive accuracy of 91.3%. The top five enriched biological pathways associated with the differentially abundant proteins included insulin growth factor functions, innate immunity, platelet degranulation, and extracellular matrix organization.

**Abstract:**

Albuminuria may precede decreases in the glomerular filtration rate (GFR) and both tests are insensitive predictors of early stages of kidney disease. Our aim was to characterise the urinary proteome in black African individuals with albuminuria and well-preserved GFR from South Africa. This case-controlled study compared the urinary proteomes of 52 normoalbuminuric (urine albumin: creatinine ratio (uACR) < 3 mg/mmol) and 56 albuminuric (uACR ≥ 3 mg/mmol) adults of black African ethnicity. Urine proteins were precipitated, reduced, alkylated, digested, and analysed using an Evosep One LC (Evosep Biosystems, Odense, Denmark) coupled to a Sciex 5600 Triple-TOF (Sciex, Framingham, MA, USA) in data-independent acquisition mode. The data were searched on Spectronaut^TM^ 15. Differentially abundant proteins (DAPs) were filtered to include those with a ≥2.25-fold change and a false discovery rate ≤ 1%. Receiver–operating characteristic curves were used to assess the discriminating abilities of proteins of interest. Pathway analysis was performed using Enrichr software. As expected, the albuminuric group had higher uACR (7.9 vs. 0.55 mg/mmol, *p* < 0.001). The median eGFR (mL/min/1.73 m^2^) showed no difference between the groups (111 vs. 114, *p* = 0.707). We identified 80 DAPs in the albuminuria group compared to the normoalbuminuria group, of which 59 proteins were increased while 21 proteins were decreased in abundance. We found 12 urinary proteins with an AUC > 0.8 and a *p* < 0.001 in the multivariate analysis. Furthermore, an 80-protein model was developed that showed a high AUC ˃ 0.907 and a predictive accuracy of 91.3% between the two groups. Pathway analysis found that the DAPs were involved in insulin growth factor (IGF) functions, innate immunity, platelet degranulation, and extracellular matrix organization. In albuminuric individuals with a well-preserved eGFR, pathways involved in preventing the release and uptake of IGF by insulin growth factor binding protein were significantly enriched. These proteins are indicative of a homeostatic imbalance in a variety of cellular processes underlying renal dysfunction and are implicated in chronic kidney disease.

## 1. Introduction

Chronic kidney disease (CKD) affects between 9.1 and 13.4% of people and is one of the main causes of death worldwide [1,2,3]. CKD often progresses silently and laboratory tests are used to diagnose, stage, and monitor its improvement or progression [2].

According to the recently published Kidney Disease: Improving Global Outcomes (KDIGO) clinical practice guidelines, CKD is defined as the presence of a reduced glomerular filtration rate (GFR) < 60 mL/min/1.73 m^2^ or albuminuria, defined as a urine albumin:creatinine ratio (uACR) ≥ 3 mg/mmol) for more than 3 months [4]. However, these markers have well-known limitations. Approximately 50% of the kidney function is compromised when the eGFR falls below 60 mL/min/1.73 m^2^, thereby making patients vulnerable to a risk of developing drug toxicity and metabolic and endocrine abnormalities linked with declining kidney function [5]. As the GFR falls below this threshold, there is a marked increase in the risk of kidney failure, cardiovascular complications, and all-cause mortality [6,7].

Damage to the glomerular filtration barrier leads to impaired size and charge selectivity, causing albumin leakage [8]. Regardless of the cause of albumin leakage, data suggest that albuminuria is not only a marker of kidney damage but also exerts a direct toxic effect on renal tubules, resulting in progressive loss of function [9]. Some studies show a moderate diagnostic accuracy of albuminuria for CKD, with a lack of sensitivity at low levels [10,11,12]. Unfortunately, albuminuria often manifests after substantial kidney damage has already occurred [13,14].

The identification of proteomic markers that are dysregulated in albuminuria could allow the non-invasive evaluation of individuals at risk of CKD [15]. Early diagnosis and understanding of the pathophysiological mechanism behind CKD progression remains a critical area of research due to the limitations of the traditional markers. Proteomic-based biomarkers have the potential to overcome the limitations of current markers for the diagnosis of CKD. Analysis of the urinary proteome is attractive, as urine can be collected non-invasively and in large quantities, and may reflect a biological process specific to the kidney [16]. Proteomic-based analysis showed a higher diagnostic precision for CKD and a better ability to identify patients with a rapid kidney function decline beyond the detection of albuminuria in a Norwegian population [17]. In a cross-sectional study in the Chinese population, urinary proteome profiling was able to differentiate uncomplicated diabetes from diabetic nephropathy at different stages [18].

The discovery of proteins related to CKD has the potential not only to improve understanding of its pathophysiology and progression, but also to establish new markers for the diagnosis of CKD [19]. Before such markers can be applied in clinical settings, multi-ethnic studies should be conducted, especially in African populations, which are largely unrepresented by the current literature. Limited or no studies have been published in South Africa (SA) and Africa. A recent study from SA demonstrated the ability of urinary peptidomics to discriminate normotensive from hypertensive individuals using a classifier that consists of 20 urine peptides [20]. Another study in SA, using urinary proteomics coupled with machine learning tools, showed that proteomics-based markers could classify participants with hypertensive-associated albuminuria from healthy controls [21]. Identifying early markers could significantly streamline efforts to delay progression towards end-stage renal disease, especially in low-and middle-income countries experiencing a high burden of CKD with limited resources. Therefore, proteins associated with albuminuria can serve as early-stage diagnostic markers for CKD. In this pilot study, the aim was to characterise the urinary proteome in individuals with albuminuria with a well-preserved eGFR in a South African cohort.

## 2. Materials and Methods

### 2.1. Ethics Statement

This study received approval (certificate number: M210128) from the Faculty of Health Sciences Human Research Ethics Committee (Medical) at the University of the Witwatersrand, Johannesburg, South Africa.

### 2.2. The Study Population

Our study was a sub-study of the African Research on Kidney Disease (ARK) study, which had a population-based sample size of 2021 adults aged 20–80 years of self-identified black ethnicity from rural Bushbuckridge in the Mpumalanga province, SA. The ARK study aimed to determine the prevalence of CKD and its associated risk factors. Detailed methods for the ARK study have already been published [22]. All demographic and clinical information collected from this study was captured in REDCap (V.22.3.4, Vanderbilt University, Nashville, TN, USA) [23]. For the present study, we included urine samples from 108 participants from the ARK study. We included participants with confirmed albuminuria (cases) on repeated testing and normoalbuminuria (controls) groups, irrespective of their estimated GFR and regardless of CKD risk factors. Cases and controls were age- and sex-matched, with a 5-year range applied if the exact match was not achieved. Cases were defined as having albuminuria (uACR ≥ 3 mg/mmol), while controls were defined as normoalbuminuria (uACR < 3 mg/mmol), in line with the KDIGO guidelines [5]. In the original study, apriori, participants were classified as being hypertensive (SBP ≥ 140 mm Hg and/or diastolic blood pressure [DBP] ≥ 90 mm Hg) based on the 7th Report of the Joint National Committee on Prevention, Detection, Evaluation, and Treatment of High Blood Pressure [24,25]. Diabetes was defined as a non-fasting glucose ≥ 11.1 mmol/L and human immunodeficiency virus (HIV) status was recorded as positive if participant knew their HIV status as positive. Voluntary HIV testing was offered to participants with unknown status or who previously tested negative [25].

### 2.3. Clinical Laboratory Tests

Serum creatinine was measured using Jaffe’s method, traceable to isotope dilution mass spectrometry, while urine albumin was measured using an immunoturbidimetry method on a Roche analyzer (Roche Diagnostics, Mannheim, Germany). The eGFR was calculated using the Chronic Kidney Disease Epidemiology Collaboration (CKD-EPI) equation 2009 from serum creatinine without race correction, as suggested by the previous literature from African populations [26].

### 2.4. Urine Protein Extraction

Urinary proteome preparation was performed using an in-house method as previously described [21]. Briefly, four times (1600 μL), ice-cold 80% acetone was added to 400 μL of urine in 2 mL protein Lo-Bind tubes (Eppendorf, Hamburg, Germany). After incubating at −20 °C for 1 h, samples were centrifuged at 12,000× *g* for 30 min. The supernatant was removed, and the pellet dried using a 70 °C heating block (AccuBlock Digital dry bath, Labnet International Inc., Edison, NJ, USA) for 1 min. Pellets were resuspended in 100 μL 2% sodium dodecyl sulfate (SDS) and sonicated for 5 min. Proteins were reduced with 1 μL of 1M dithiothreitol (DTT) at 70 °C for 15 min, and thereafter transferred to a 40 °C heating block for an additional 15 min, followed by alkylation with 6 μL of 500 mM iodoacetamide (IAA) for 30 min at room temperature (RT) in the dark. Proteins were on-bead digested using MagResyn^TM^ HILIC microparticles (ReSyn Biosciences, Edenvale, South Africa) using an automated workflow in a KingFisher^TM^ Duo (Thermo Fisher Scientific, Rockford, IL, USA), as previously described [21]. The peptides were dried with a CentriVap vacuum concentrator (Labconco, Kansas City, MO, USA) overnight, resuspended in 40 μL of 2% acetonitrile/0.2% formic acid, and stored at −80 °C until LC–MS/MS analysis. Peptide quantification was done using the Pierce™ Quantitative Colorimetric Peptide Assay (Thermo Fisher Scientific, Waltham, MA, USA) as per the manufacturer’s instructions. A pooled sample from 10 patient urine samples was prepared and analysed alongside individual samples as system quality control (QC) for this study. In addition, a commercial Hela digest system suitability QC was analysed.

Tryptic peptides were analysed using an Evosep One LC system (Evosep Biosystems, Odense, Denmark) interfaced to a SCIEX TripleTOF 5600 tandem mass spectrometer (Sciex, Framingham, MA, USA) in data-independent acquisition (DIA) mode. An Evosep performance column (EV1112, 15 cm × 75 μm, 1.9 μm) was used for the Whisper 40SPD method. The Nanospray 3 source settings were as follows: CUR–20, GS1–30, ISVF–2900. Data were acquired using 48 MS/MS scans of overlapping sequential precursor isolation windows (variable *m*/*z* isolation width, 1 *m*/*z* overlap, high sensitivity mode), with a precursor MS scan for each cycle. The accumulation time was 50 ms for the MS1 scan (from 400 to 1100 *m*/*z*) and 30 ms for each product ion scan (from 200 to 1800 *m*/*z*) for a 1.53 s cycle time.

A spectral library was built in Spectronaut^TM^ 15 software (Biognosys, Schlieren, Switzerland). Default settings were used with minor adjustments as follows: segmented regression was used to determine the normalized retention time (iRT) in each run; iRTs were calculated as median for all runs; the digestion rule was set as “Trypsin” and modified peptides were allowed; fragment ions between 300 and 1800 *m*/*z* and peptides more than three amino acids were considered; peptides with a minimum of 3 and maximum of 6 (most intense) fragment ions were accepted. This study-specific spectral library was combined with an in-house-generated urinary proteome spectral library (using in Spectronaut™ “Search Archives” feature). Raw (.wiff) data files were analysed using Spectronaut™ 15 with default settings targeted analysis. These default settings included: dynamic iRT retention time prediction with a correction factor for window 1; mass calibration was set to local; decoy method was set as scrambled; the false discovery rate (FDR), according to the mProphet approach [27], was set at 1% on the precursor, peptide, and protein group levels; protein inference was set to “default”, based on the ID picker algorithm [28]; and global cross-run normalization on the median was selected. The final urinary proteome spectral library (peptides—20,616, protein groups—2604) was used as a reference for targeted data extraction.

### 2.5. Retrospective Power Analysis

The appropriate fold-change cut-off for group comparisons was determined by performing retrospective power analysis using the MSstats package (Northeastern University, MSstat 4.4.1, Bioconductor version: release 3.15, R v4.2.0). A group comparison was performed to compare the protein changes between cases and controls. To achieve desirable statistical power, parameters were: false discovery rate (FDR) = 0.01 (1%); *n* = minimum sample size for each comparison was 52. At power = 80%, proteins that showed a fold-change ≥ 2.25 were statistically significant.

### 2.6. Data Analysis and Pathway Analysis

The demographic and biochemical characteristics were analysed using STATA 17 SE (Stata Corp, College Station, TX, USA) and GraphPad Prism 10 (GraphPad Software, San Diego, CA, USA). Continuous and categorical variables were analysed by Mann–Whitney U test and chi-square (χ^2^), respectively. The continuous variables are expressed as medians (interquartile ranges), and categorical variables as proportions. A *p*-value < 0.05 was considered statistically significant. A volcano plot was generated using online data analysis and visualization tool http://www.bioinformatics.com.cn/srplot (accessed on the 15 January 2024) and principal component analysis (PCA) was performed on Metaboanalyst v.5.0 (https://www.metaboanalysts.ca/). Differentially abundant proteins between cases and controls were calculated by a two-sided *t*-test, with a minimum fold change ≥ 2.25 and *p*-values adjusted for multiple testing by FDR at 1% from Spectronaut™ 15. Receiver–operating characteristic (ROC) curves were generated using GraphPad Prism on all differentially abundant proteins (DAPs) to identify putative marker proteins that differentiate albuminuria patients from those without. Data were log-transformed, and interquartile range filtering and zero imputation strategy was used. Normalization by median was performed. Multivariate ROC exploratory analysis was generated using the Monte Carlo cross-validation algorithm to identify proteins or models with high sensitivity and specificity. In each validation algorithm, two-thirds of the participant data were used to build a model, while the remaining one-third was used to validate the model. This process was replicated several times to calculate the performance and confidence intervals [29]. The results of the ROC curve analysis are reported as the area under the curve (AUC) [30]. Functional enrichment analysis on DAPs was performed using a free online tool, Enrichr/Enrichr-KG (https://maayanlab.cloud/Enrichr, accessed on 20 January 2024), with the Reactome 2022 human database [31,32]. The top enriched pathways (*p* < 0.05) were selected.

## 3. Results

### 3.1. Clinical and Demographic Characteristics of Patients

The baseline characteristics and biochemical variables of the study population are summarized in Table 1. From the initial 116 participants selected for analysis, eight were excluded post-data acquisition due to poor chromatographic separation or low peptide recovery. The median age of the sample (*n* = 108) was 42 years, with 57% being female. There were significantly more HIV-positive individuals (*p* = 0.033) in the albuminuric group, and the uACR was significantly higher in this group (7.9 mg/mmol vs. 0.6 mg/mmol, *p* < 0.001).

### 3.2. Performance of Study-Specific System Suitability-Quality Control

The performance of study-specific and commercial QCs (Hela) is shown in Appendix A. The coefficient of variation (CV) at the protein-group level was 15% in the Hela digest system suitability assessment and 14% for the urine pool over 5 days of analysis. The CV at the peptide level was 18.2% and 17.5% for the Hela and urine pool tests, respectively. The counts for the protein, peptide, and precursors remained consistent throughout the data acquisition process (Appendix A).

### 3.3. Differentially Abundant Proteins between Cases and Controls

The differentially abundant proteins (FDR ≤ 1%, fold-change ≥ 2.25) are shown in Figure 1. Eighty urinary proteins (Appendix A) were differentially abundant between the groups. Among them, 59 urine proteins (74%) showed higher abundance, while 21 proteins (26%) showed lower abundance in the cases compared to controls. PCA revealed modest clustering between the groups (Figure 2). The top 10 DAPs that were in higher abundance (based on q values) in the cases comprised of alpha-1-antitrypsin (SERPINA1), albumin, afamin, antithrombin III (SERPINC1), vitamin-D binding protein, alpha-1B glycoprotein, beta-ala His dipeptidase, myoglobin, fibrinogen beta chain, and apolipoprotein A-1 (Appendix A). The top 10 DAPs found in lower abundance in the cases (higher abundance in the control group) included polymeric immunoglobulin receptor, collagen alpha-1(VI) chain, mucin-1, collagen alpha-1(XV), inducible costimulatory (ICOS) ligand, collagen alpha-2(IV) chain, uromodulin, intercellular adhesion molecule, osteopontin, and platelet endothelial cell adhesion (Appendix A).

### 3.4. Potential Markers for Albuminuria and Normoalbuminuria Classification

ROC curve analysis was performed to evaluate the diagnostic potentials of differentially abundant proteins. Using an AUC > 0.8 and a *p*-valve < 0.05 as criteria for diagnostic potential, 12 urine proteins were identified as potential markers for albuminuria (Figure 1, Figure 3 and Figure 4, Appendix A). Among the 12 urine proteins, SERPINA1, albumin, SERPINC1, and afamin exhibited AUCs greater than 0.9 (AUC = 0.951, 0.936, 0.920, 0.903, respectively). Among the 12 urine proteins with the highest AUCs, seven proteins were higher in abundance (SERPINA1, alpha-1B glycoprotein, albumin, afamin, SERPINC1, myoglobin, and immunoglobulin lambda variable 3–9), while five proteins (polymeric immunoglobulin receptor, collagen alpha-1(VI) chain, mucin-1, uromodulin, and inducible costimulatory ligand) were lower in abundance, among the cases compared to controls. Finally, a multivariate ROC curve analysis was performed using a combination of proteins selected through logistic regression analysis, as shown in Figure 5a. All six urine protein models had an AUC > 0.8. However, the first three models had wider confidence intervals than models 3 and 4 (Figure 5a). The highest AUC of 0.907 (confidence interval (CI): 0.836–0.954) was achieved with the sixth model, comprising 80 urine proteins (Appendix A). Among all models, the sixth model had the highest predictive accuracy of 91.3% in classifying cases, even though the increase was modest compared to the fourth and fifth urine protein models (Figure 5b).

### 3.5. Functional Enrichment Analysis of Differentially Abundant Proteins

Pathway analysis was performed by querying the Reactome library, within the Enrichr free online tool, to identify enriched pathways in which differentially abundant proteins are involved. The significantly enriched functional pathways based on the FDR (≤1%) and *p*-value < 0.05 are illustrated in Appendix A. The top five highly enriched pathways for all DAPs included: the innate immune system, the regulation of insulin-like growth factor (IGF) transport and uptake by insulin-like growth factor binding proteins (IGFBPs), platelet degranulation, the response to elevated cytosolic Ca^++^, and haemostasis (Appendix A). In the biological function (GO) analysis for urinary proteins that showed higher abundance in the cases, the top five highly enriched terms were involved in the following biological processes (Figure 6): regulated exocytosis, platelet degranulation, high-density particle remodelling, the hydrogen peroxide catabolic process, and the hydrogen peroxide metabolic process. Some proteins that showed lower abundance among the cases were enriched in processes such as extracellular structure organization, extracellular matrix organization, and extracellular encapsulating structure organization, as well as in homophilic cell-adhesion molecules (Figure 7, Appendix A).

## 4. Discussion

In this study which profiled the urinary proteome in Black African adults with albuminuria and well-preserved eGFR, we found 80 differentially expressed proteins when comparing the albuminuric cases to non-albuminuric controls. Using multivariable analysis, we identified a combination of 80 proteins (AUC = 0.907) that outperformed other models with a predictive accuracy of 91.3%, suggesting that a combination of markers may offer added clinical utility compared to a single marker [33]. In this model, five proteins (ALB, AFM, PIGR, A1AT, ANT3) were found to be differentially expressed, similarly to a 20-protein classifier identified in a South African study of hypertension-associated albuminuria [21] and in previous studies [34,35,36], suggesting generalisability of these markers to different populations.

We observed a higher proportion of HIV-positive adults in the albuminuria group compared to controls. HIV-infected individuals are likely to have kidney damage due to direct virus-induced kidney injury or secondary antiretroviral therapy (ART) [37]. It has previously been shown that the prevalence of albuminuria significantly increases, ranging between 34 and 68%, in HIV-infected individuals compared to HIV-uninfected controls [38,39].

Most DAPs were involved in pathways involving the innate immune system, the regulation of IGF transport and uptake by IGFBPs, platelet degranulation, the response to elevated cytosolic Ca^++^, and haemostasis. Dysregulation of the immune system and inflammatory events have been postulated as key mechanisms in the pathogenesis of CKD [40]. Kidney injury from any cause eventually evolves into CKD in the presence of unresolved inflammation. Following kidney injury, damage-associated molecules initiate inflammatory responses such as proinflammatory cytokine activation, and the subsequent infiltration of immune cells, such as neutrophils, macrophages, and natural killer cells [41,42]. The role of IGFBPs in regulating transcription, cell migration induction, and cell-cycle arrest, and preventing tissue renewal, triggering apoptosis, is linked to the development of fibrosis in kidney disease [43]. Similarly to the findings of this study, evidence indicates that IGFBPs are upregulated in various kidney diseases [44,45,46]. One mechanism of action of IGFBPs is through binding to insulin-like growth factor (IGF) [47]. IGF plays a significant role in enhancing kidney injury repair [47]. The upregulation of pathways that prevent the release and actions of IGF is implicated in the kidney in CKD [48]. The administration of sodium-glucose co-transporter 2 (SGLT2) inhibitors has been shown to abolish the effect of IGFBP, especially IGFBP-7 [49]. This effect is believed to be the main cause of the kidney-protective effect of SGLT2 inhibitors, independent of the hypoglycaemic effect, in the treatment of type 2 diabetes [50].

We identified nine proteins that were linked to the activation of platelet degranulation pathways. Among these, seven were increased in the albuminuric group. Participants with CKD demonstrated increased platelet activation [51]. In addition to their role in thrombosis, platelets appear to modulate inflammatory processes that are integral to the pathophysiology of CKD and its progression [39,47,48]. Importantly, other studies have conflicting results of reduced platelet activation or unchanged platelet activation. However, these studies mainly included patients on chronic dialysis and with uraemic syndrome. The haemostasis and platelet degranulation pathways often occur together, which could present an opportunity for treatment targets [52,53]. Furthermore, elevated cytosolic calcium leads to the activation of protein kinase C (PKC), a hallmark of the response to platelet degranulation. It is not surprising that pathways to the cytosolic calcium response pathway are also highly enriched alongside platelet degranulation and haemostasis [40]. Coupled with inflammatory pathways, CKD is a highly oxidative state [54]. High levels of oxidative stress have already been detected in the early stages of CKD, and increase with the progression to end-stage renal disease [55,56]. These observations are supported by highly enriched biological function processes associated with hydrogen peroxidase catabolic and metabolic processes among the DAPs found in this study.

In keeping with the growing evidence, DAPs that were decreased in the albuminuria group were enriched in processes such as extracellular structure organization, extracellular matrix organization, and extracellular encapsulating structure organization, as well as in homophilic cell-adhesion molecules. It is well-accepted that the accumulation of extracellular matrix disrupts the cellular organisation and function of the kidney. An increase in extracellular structure matrix proteins is a key finding in renal fibrosis [57,58]. Decreased urinary uromodulin, osteopontin, and collagen fragments have all been implicated in fibrosis-promoting processes in CKD [17,59]. There is mounting evidence that collagen fragments, used as biomarkers, reflect structural changes in the renal extracellular matrix (ECM) and provide insight into the progression of fibrosis [60]. We observed an increase in urinary collagen alpha-1(III) in the albuminuria group while other collagen fragments were low (i.e., collagen alpha-1(VI), collagen alpha-1(XV), collagen alpha-2(IV)). Different types of collagen are involved in maintaining the integrity and function of the renal structure. The upregulation of collagen III is regarded as an early event in renal fibrosis and is associated with adverse CKD outcomes [61]. Elevated levels of collagen alpha-1(III) in the urine could indicate an increased turnover or breakdown of collagen in the renal interstitium, possibly as a result of kidney injury or inflammation [62]. While collagen type IV (COL VI), a major component of basement membranes, is crucial for maintaining the structural integrity of the glomeruli, studies have shown high urinary COL VI levels, especially in diabetic patients, compared to healthy controls [63,64]. The reduced urinary COL VI found in this study may partly be explained by a shift in a deposition associated with the severity of the disease in favour of degradation. This may be due to reduced matrix metalloproteases activity and the increased cross-linking of collagen fibres that render them resistant to proteolytic processes [60,65,66].

Similarly, other studies found a lower abundance of urine UMOD in the cases than in controls [67,68]. The reduced levels are believed to be secondary to tubular dysfunction [69]. Similarly, in the South African study of hypertensive-associated albuminuria, UMOD was found in lower abundance [21]. UMOD, a mucoprotein synthesized by the thick ascending loop of Henle, has been shown to be a marker of renal reserve and plays an important role in the innate immunity of the kidney [59,70]. Low urine UMOD levels have been associated with acute kidney injury and its progression to CKD [71].

This case-control study characterized the urinary proteome in South African participants with and without albuminuria. Most of the urinary proteins identified in this study have also been found in a variety of diseases related to kidney conditions, such as glomerulonephritis, diabetic nephropathy, and hypertension, to name a few. Despite the diverse causes of kidney damage, commonalities in upregulated or downregulated pathways suggest that therapeutic targeting of these pathways may be crucial in preventing kidney progression, regardless of the underlying cause. The main limitation of this study was the analysis of a single cohort. The findings need to be verified in a larger and more diverse cohort of patients to ensure applicability to a wider population.

## 5. Conclusions

Pathway analysis suggested significant enrichment in innate immune pathways, platelet degranulation, IGF, and IGFBPs, as well as in haemostasis. Urine proteins that were in lower abundance were involved in pathways associated with the extracellular matrix. This information can be utilized to discern potential molecular mechanisms underlying kidney damage and its progression to severe stages of chronic kidney disease (CKD). Additionally, urine proteins such as SERPINA1, Alb, SERPINC1, AFM, PIGR, A1BG, COL6A1, MYG, LV39, MUC1, ICOSLG, and UMOD had the highest discriminating abilities between cases and controls. Our study identified an 80-protein model that was able to classify albuminuria from normoalbuminuria with well-preserved GFR with a predictive accuracy of 91.3%.

## Figures and Tables

**Figure 1 biology-13-00680-f001:**
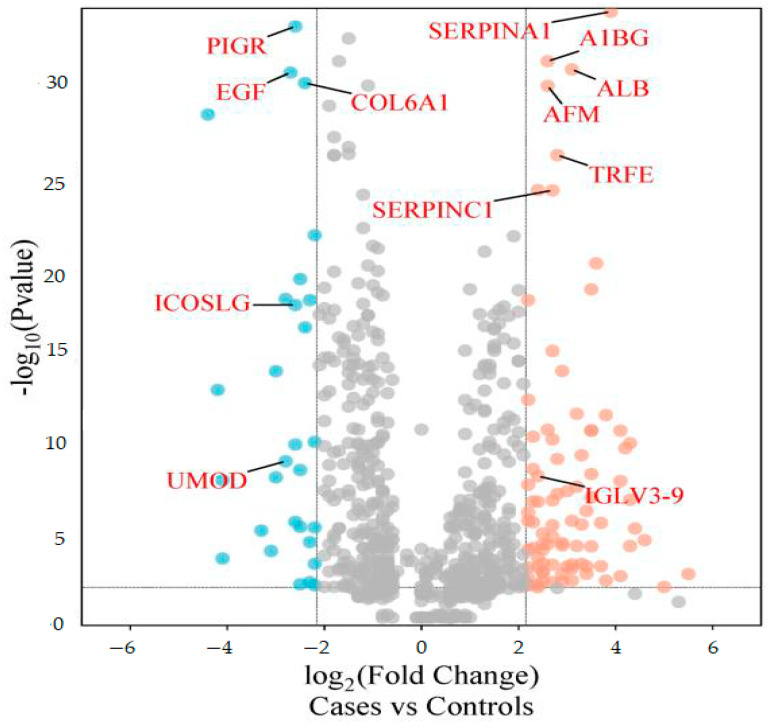
Volcano plot showing differentially abundant proteins. Candidate protein markers (orange dots indicate proteins with higher abundance in the cases and blue dots indicate proteins with lower abundance in the cases) with a cut-off of minimum fold change ≥ 2.25 and maximum 0.01% (1% FDR).

**Figure 2 biology-13-00680-f002:**
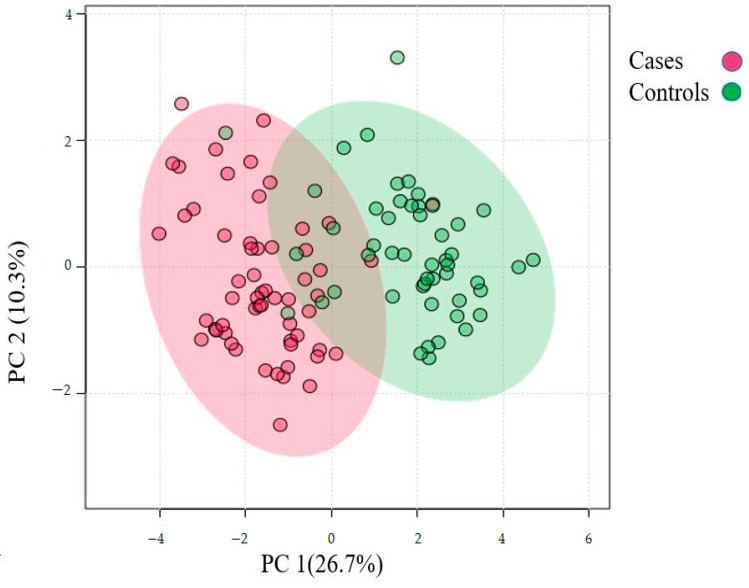
Principal component analysis of the urinary proteome of albuminuria (red) and normoalbuminuria (green).

**Figure 3 biology-13-00680-f003:**
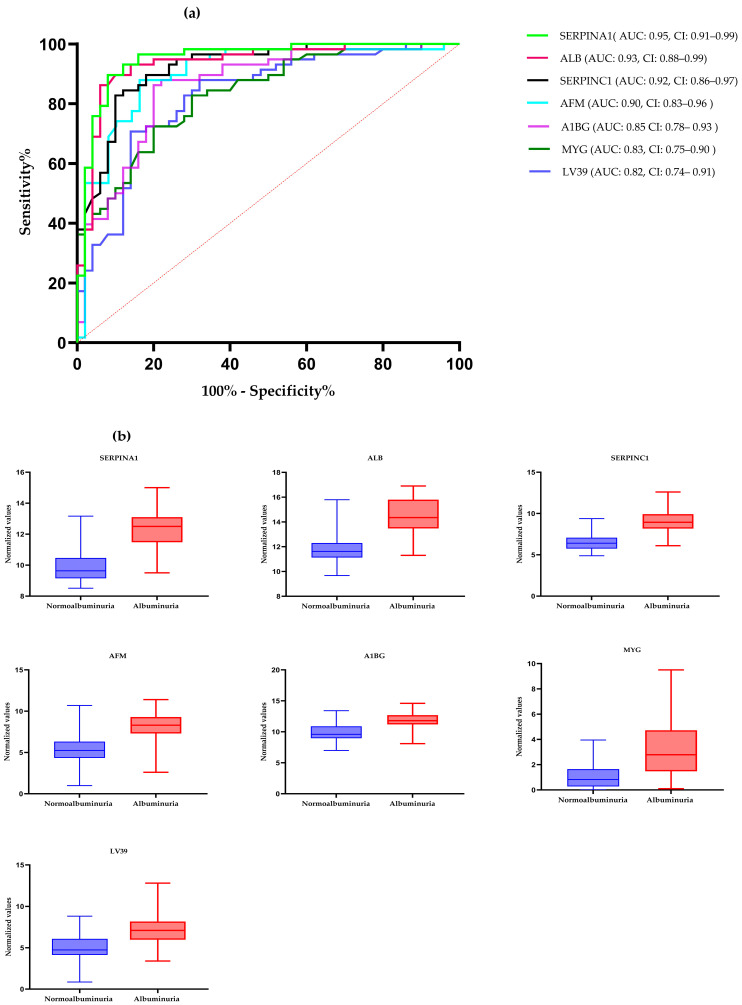
(**a**) The receiver–operating characteristic curves of the top seven highly abundant proteins in the albuminuric group. The diagonal red dotted line reflects the performance that is no better than chance. (**b**) The box-cum whisker plots show significantly different proteins between albuminuric and normoalbuminuric patients. The box denotes interquartile ranges, and the bottom and top boundaries of boxes are the 25th and 75th percentiles, respectively. Lower and upper whiskers correspond to the 5th and 95th percentiles, respectively. A horizontal line inside the box denotes the median. *p* value < 0.001 for all proteins. SERPINA1: alpha-1 antitrypsin, ALB: albumin, SERPINC1: antithrombin III, AFM: afamin, AIBG: alpha-1B-glycoprotein, MYG: myoglobin, LV39: immunoglobulin lambda variable 3–9. AUC: area under the curve, CI: confidence interval.

**Figure 4 biology-13-00680-f004:**
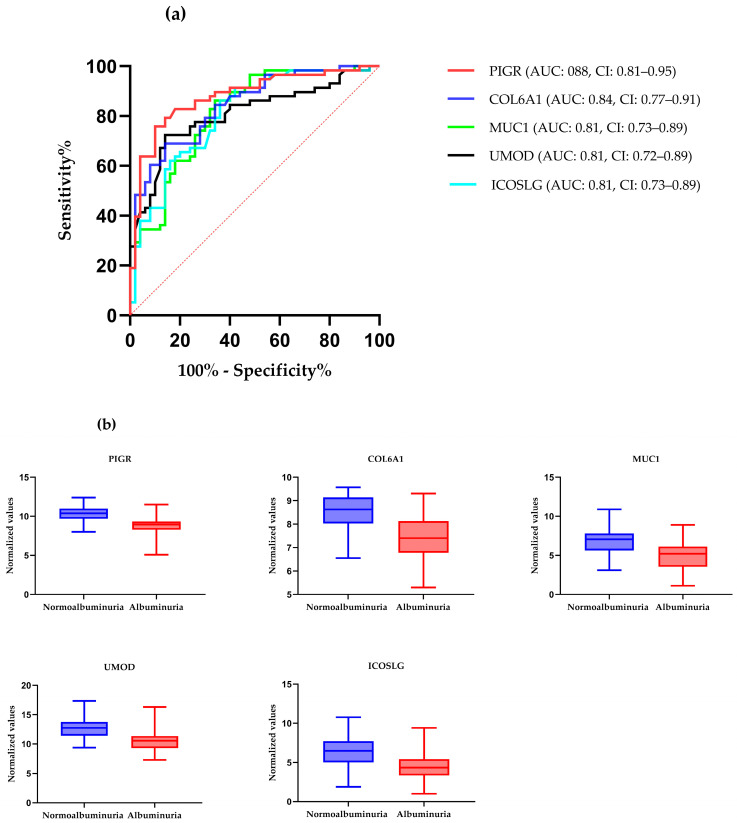
(**a**) The receiver–operating characteristic curves of the top five highly abundant proteins in the normoalbuminuric group. The diagonal red dotted line reflects the performance that is no better than chance. (**b**) The box-cum whisker plots show significantly different proteins between albuminuric and normoalbuminuric patients. The box denotes interquartile ranges, and the bottom and top boundaries of boxes are the 25th and 75th percentiles, respectively. Lower and upper whiskers correspond to the 5th and 95th percentiles, respectively. A horizontal line inside the box denotes the median. *p* value < 0.001 for all proteins, PIGR: polymeric immunoglobulin receptor, COL6A1: collagen alpha-1(VI) chain, MUC-1: mucin-1, ICOSLG: inducible costimulatory ligand, UMOD: uromodulin.

**Figure 5 biology-13-00680-f005:**
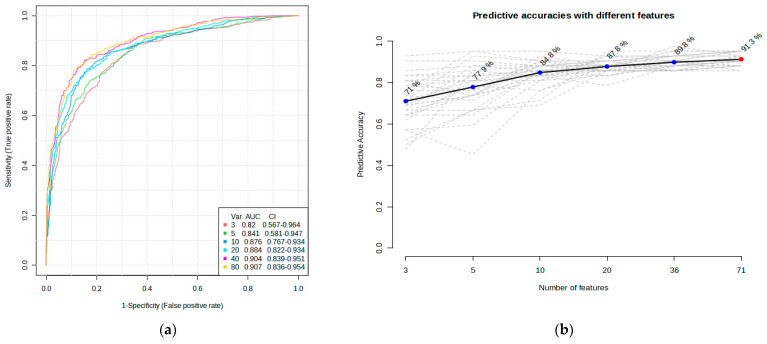
Multivariate ROC of six models and their predictive accuracies. (**a**) Performance of six models; and (**b**) predictive accuracies of all models for discriminating albuminuria from normoalbuminuria.

**Figure 6 biology-13-00680-f006:**
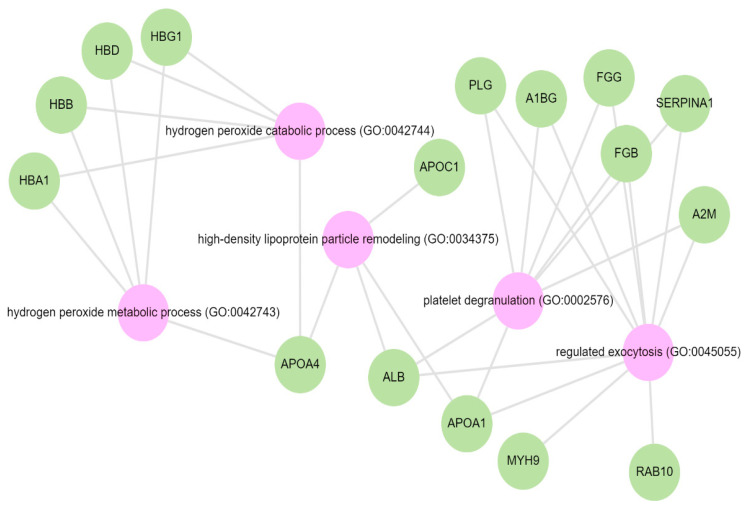
GO biological terms for differentially abundant proteins increased in the albuminuria group through Enrich-KG (Reactome library). Pink circles indicate highly enriched terms.

**Figure 7 biology-13-00680-f007:**
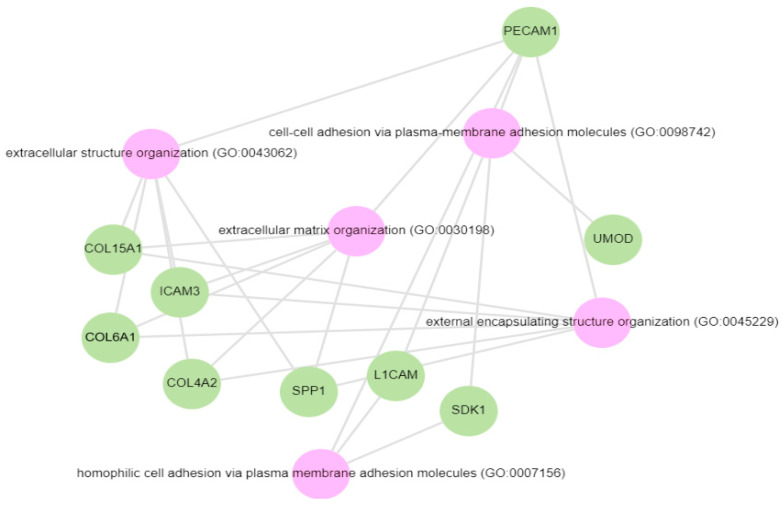
GO biological processes for proteins decreased in the albuminuria (increased in normoalbuminuria).

**Table 1 biology-13-00680-t001:** Baseline characteristics of participants.

Variable	Total	Cases (*n* = 56)	Controls (*n* = 52)	*p*-Value
Age, years	42 (30–54)	42 (30–55)	42 (31–53)	0.987
Women	61/108 (57)	32/56 (57)	29/52 (56)	0.886
BMI, kg/m^2^	25 (22–29)	25 (21–28)	25 (23–33)	0.428
Serum creatinine, µmol/L	63 (53–74)	63 (53–76)	63 (52–71)	0.550
eGFR, mL/min/1.73 m^2^	113 (95–124)	111(93–124)	114 (99–124)	0.707
uACR, mg/mmol	3.9 (0.6–8.4)	7.9 (5.5–18.5)	0.6 (0.30–1.1)	<0.001
HPT status	12/108 (11)	8/45 (18)	4/46 (9)	0.439
Diabetes status	3/108 (2.7)	3/26 (12)	0/28 (0.0)	0.064
HIV status	35/108 (32)	22/56 (39)	13/52 (25)	0.033
Smoking	17/108 (16)	8/56 (14)	9/52 (17)	0.667
Glucose, mmol/L	6.3 (5.6–7.7)	6.3 (5.7–7.7)	6.4 (5.6–7.5)	0.848

Abbreviations: BMI: body mass index, SBP: systolic blood pressure, DBP: diastolic blood pressure, HPT: hypertension, HIV: human immunodeficiency virus, eGFR: estimated glomerular filtration (CKD-EPI_creatinine_ equation 2009 without correction for race), ACR: urine albumin–creatinine ratio. For continuous variables, data were reported as median (from 25th to 75th percentile); for categorical variables, data were reported as number (*n*) of participants and percentages (%). A *p*-value < 0.05 was considered statistically significant.

## Data Availability

The mass spectrometry proteomics data have been deposited to the ProteomeXchange Consortium via the PRIDE [72] partner repository with the dataset identifier PXD054170.

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
