# Peer review of "Proteomic Analysis Identifies Dysregulated Proteins in Albuminuria: A South African Pilot Study"

_biology, 2024, doi:10.3390/biology13090680_

Round 1

Reviewer 1 Report

Comments and Suggestions for Authors

The manuscript "Proteomic analysis identifies dysregulated proteins in albuminuria: a South African pilot study" presents a very comprehensive study of urinary proteomic analysis in patients with and without albuminuria, pointing to the potential of such a method in early detection of that condition and CKD.

The text is very well written with all of the important aspects clearly stated. It very well fills the gap in the current knowledge in this area.

There are only three suggestions which should be addressed in order to increase the quality of the current version of the manuscript:

1. It would be nice if the fact that the sample had only participants of one ethnicity. Some of the previously published studies showed ethnicity differences in albuminuria incidence and thus potential differences in pathophysiological mechanisms contributing to that condition.

2. Line 125 - Please describe HIV on its first use in the manuscript.

3. Lines 194 and 195 - Shouldn't it be vice versa, i.e. that continuous variables are analyzed by Mann-Whitney U test and categorical by chi-square test? Please revise.

Other than addressing these suggestions, no further requirements are needed.

Reviewer 2 Report

Comments and Suggestions for Authors

Methods of accurate and early detection of Chronic kidney disease (CKD) and its associated albuminuria is currently not fully established. Khoza et al., in this study have generated a proteomic profiling from albuminuria patients, and by comparing with control samples they also identified protein makers differentially expressed in CKD group. Further, they built multivariate ROC models and found one of them with significant accuracy of clinic prediction. These findings will be interesting to the field for potential early diagnostics of albuminuria, especially as a complementary of the canonical glomerular filtration rate (GFR)-based measurement.

One novelty of this study as authors mentioned is these biomarkers identified from CDK patients are dysregulated independent of the decline of GFR, which is a current standard but known having the limitation for early diagnostics. Therefore, using proteomic based approaches to identify CDK biomarkers is a current research focus with potentials in clinic.  In this study, the proteomic dataset generated from South African population via approach of Data-independent acquisition (DIA) mass spectrometry could also serve a useful resource for the community. Meanwhile, I have couple suggestions to strengthen the analysis findings:  

1. In result section, authors observed a significant correlation between CDK and HIV conditions. It is known that CKD is a frequent complication of HIV infection (PMC4536633), and studies have done in mouse model suggesting that HIV genes function in glomerulosclerosis (PMID: 16988066). It could be interesting to use the same dataset generated from this study and examinate the HIV effects on CDK in South African population: whether HIV vs control group also shows differential expression of these CDK biomarkers?  In CDK group, if those HIV products can be detected and significantly enriched?

2. One suggestion/ analysis can be further done to prove the utility of this prediction model is to take advantage of proteomic results generated from other studies and test this model.

Overall, I find this study comprehensive with proper experimental design, data analysis sufficient references, and happy to recommend for a publication consideration.
